# The evolutionary path of chemosensory and flagellar macromolecular machines in *Campylobacterota*

Ran Mo[1,2,3,4☯], Siqi Zhu[1,2,3,4☯], Yuanyuan Chen[1,2,3,4], Yuqian Li[1,2,3], Yugeng Liu[1,2,3,4], Beile Gao[1,2,3]*

1 CAS Key Laboratory of Tropical Marine Bio Resources and Ecology, Guangdong Key Laboratory of Marine Materia Medica, Innovation Academy of South China Sea Ecology and Environmental Engineering, South China Sea Institute of Oceanology, Chinese Academy of Sciences, Guangzhou, China, 2 Tropical Marine Biological Research Station in Hainan, Chinese Academy of Sciences and Hainan Key Laboratory of Tropical Marine Biotechnology, Sanya, China, 3 Southern Marine Science and Engineering Guangdong Laboratory (Guangzhou), Guangzhou, China, 4 University of Chinese Academy of Sciences, Beijing, China

☯ These authors contributed equally to this work.
* gaob@scsio.ac.cn

**Data Availability Statement:** All relevant data are within the manuscript and its Supporting Information files.

**Funding:** This research was supported by National Natural Science Foundation of China (31870064 to

## Abstract

The evolution of macromolecular complex is a fundamental biological question, which is related to the origin of life and also guides our practice in synthetic biology. The chemosensory system is one of the complex structures that evolved very early in bacteria and displays enormous diversity and complexity in terms of composition and array structure in modern species. However, how the diversity and complexity of the chemosensory system evolved remains unclear. Here, using the *Campylobacterota* phylum with a robust "eco-evo" framework, we investigated the co-evolution of the chemosensory system and one of its important signaling outputs, flagellar machinery. Our analyses show that substantial flagellar gene alterations will lead to switch of its primary chemosensory class from one to another, or result in a hybrid of two classes. Unexpectedly, we discovered that the high-torque generating flagellar motor structure of *Campylobacter jejuni* and *Helicobacter pylori* likely evolved in the last common ancestor of the *Campylobacterota* phylum. Later lineages that experienced significant flagellar alterations lost some key components of complex scaffolding structures, thus derived simpler structures than their ancestor. Overall, this study revealed the co-evolutionary path of the chemosensory system and flagellar system, and highlights that the evolution of flagellar structural complexity requires more investigation in the *Bacteria* domain based on a resolved phylogenetic framework, with no assumptions on the evolutionary direction.

## Author summary

Chemosensory system is the most complicated signal transduction system in bacteria with great diversity in both composition and structural organization across species. One of its important signaling output is flagellar motility driven by a propeller, which is made of

B.G.), Key Special Project for Introduced Talents Team of Southern Marine Science and Engineering Guangdong Laboratory (Guangzhou) (GML2019ZD0407 to B.G.), Strategic Priority Research Program of the Chinese Academy of Sciences (XDA19060301 to B.G.), and Innovation Academy of South China Sea Ecology and Environmental Engineering, Chinese Academy of Sciences (NO.ISEE2021ZD03 to B.G.).The funders had no role in study design, data collection and analysis, decision to publish, or preparation of the manuscript.

**Competing interests:** The authors have declared that no competing interests exist.

dozens of proteins and shows considerable variation and complexity surrounding the core motor structure in different species. The evolution of both chemosensory system and flagellum are important biological questions but remain obscure. Here, we carefully examined the evolutionary paths of chemosensory system and flagellar structure in a bacterial phylum, providing detailed molecular evidences for their co-evolution. Our study provides a paradigm to study the evolution of macromolecular complexes based on robust bacterial phylogeny and co-evolved systems/components in genome context.

## Introduction

The evolution of macromolecular machines is a central biological issue. Answers to this question are related to the origin of life, the formation and functional divergence of complex structures, and also provide design principles for bottom-up construction in synthetic biology. Among all the macromolecular machines in single-celled microbes, chemosensory system is one of the highly organized ultra-structures that can be visualized using electron microscopy [1]. Chemosensory system is employed by almost all motile bacteria and archaea to navigate motility or regulate other cellular processes such as cell division, lifecycle, and virulence [2–4]. To make decisions regarding such important behaviors, this system is undoubtedly the most complicated signal transduction system in bacteria and archaea [5].

The chemosensory system originated from a simpler two-component system but evolved into a multi-component complex to achieve multiple-input signal integration and tunable information processing [6]. The core components of the chemosensory pathway include chemoreceptors (also called **m**ethyl-accepting **c**hemotaxis **p**roteins, [MCPs]), CheA kinase and CheW adaptor [7, 8]. Thousands of these proteins assemble into large arrays within cells, either transmembrane or cytoplasmic [9, 10]. Upon stimulation by ligands or other environmental factors, chemoreceptors affect the autophosphorylation of CheA, which can pass the phosphoryl group to the response regulator CheY [11]. Phosphorylated CheY can interact with signaling outputs such as the flagellar motor switch protein to control the direction of flagellar rotation [12, 13]. In addition, other auxiliary components such as CheBR and CheZ/CheC/CheX phosphatases fine-tune the signaling process [14].

Although the chemosensory system has the same signaling mechanism as that described above, a large-scale comparative genomic study revealed a remarkable diversity of different chemosensory pathways [2]. Based on phylogenomic markers, the chemosensory system is sorted into 19 classes, including one Tfp class involved in type IV pili-based motility, one ACF class with alternative (non-motility) cellular functions, and 17 F classes (F1—F17) that control flagellar motility in some lineages but not necessarily in all cases [2]. These 19 classes show distinctive gene order and domain architecture of their components, with different auxiliary component(s) and specific types of chemoreceptors (see updated information of ref. 2 in S1 Table). Notably, more than half of the motile bacteria have more than one chemosensory class within one cell [2]. For example, *Pseudomonas aeruginosa* PAO1 has four classes (F6, F7, ACF, and Tfp); *Azospirillum* sp. B510 genome encodes six classes (F5, F7, F8, F9 and 2 ACFs); and *Cystobacter fuscus* DSM2262 contains 12 classes (F1, F10, 3 F8 and 7 ACFs). In contrast, the model organisms *E.coli* and *Bacillus subtilis* have only one chemosensory class. Furthermore, recent developments in electron cryotomography have revealed great structural variations in chemosensory arrays made of individual classes in terms of their protein composition and stoichiometry, architecture, localization, and functional variability (S1 Table) [15]. Underneath diversity, why and how did chemosensory system evolve into different classes? This

fundamental question remains elusive, but has been approached by researchers through the study of more bacterial species with distinctive or multiple classes.

Electron cryotomography studies coupled with bioinformatics and genetic analyses can link structural features with compositional differences in chemosensory classes, facilitating our understanding of their functional divergence. For example, the F2 class is restricted to *Spirochaetota* and Muok *et al.* demonstrated that F2 arrays have atypical extended structures with unusual protein domain organization to accommodate the high membrane curvature of *Spirochaetota* [16]. *Vibrio cholerae* has three classes, F6, F7, and F9, and recent studies have shown that only F6 is responsible for chemotactic motility, whereas F7 and F9 are induced to form arrays under stress conditions although their signal outputs remain unknown [17, 18]. *Myxococcus xanthus* has eight classes and one of the ACFs (Frz) that regulates both A- and S-motility can form multiple small arrays at the nucleoid, displaying unusual cellular localization in addition to a complicated output switch [19, 20]. In addition, tracking the evolutionary history of chemosensory system can also help elucidate the diversity and complexity we see today. The Frz pathway in *M. xanthus* has been suggested to have evolved from a simpler ancestral form that only originally controlled S-motility, shedding light on the origin of the branched signaling pathway [19]. Surprisingly, the paradigm chemosensory system in *E. coli* was recently identified as a hybrid of components from F6 and F7 classes by tracing the evolution of F classes in γ-proteobacteria [21].

Recently, we proposed that the *Campylobacterota* phylum (previously the ε-proteobacteria class) is ideal for tracking the evolutionary history of biological systems because it can offer a temporal direction of evolution correlated with ecological diversification [22]. The ancestor of this phylum was an anaerobic chemolithotroph from deep-sea hydrothermal vent; later lineages underwent niche expansion with broad ecological distribution and heterotrophic lifestyle; and finally, some lineages developed into host-associated commensals or pathogens [22]. Based on the ecological and evolutionary framework of *Campylobacterota*, our genomic analyses of the chemosensory system revealed that the F3 class was vertically inherited from the ancestor of this phylum, whereas additional classes such as F7, F8, and F9 were acquired by horizontal gene transfer (HGT) during lineage niche expansion, but subsequently lost in host-associated lineages [22]. We concluded that the chemosensory system evolves in the manner of "add, reduce, but rarely stir", highlighting the dynamic gain and loss of chemosensory classes during niche adaptation of species and particularly the rare "mix and match" events of components among classes [22]. Our analyses provide important information on the evolutionary mode of the chemosensory system; however, the determinants of both the conservation of the F3 class and the evolutionary changes of other F classes in this phylum have not been studied.

A complete evolutionary scenario of the chemosensory system is indispensable with its cellular functions. However, one of the conundrums in studies of species with multiple chemosensory classes is to predict and test their regulated targets, which cannot be easily done, except for very few model organisms such as *P. aeruginosa* [23]. The chemosensory system is functionally characterized in two important pathogens of the *Campylobacterota* phylum, namely *Helicobacter pylori* and *Campylobacter jejuni*, due to its important role in pathogenesis [24–27]. Both *H. pylori* and *C. jejuni* contain only one F3 class, and this class controls flagellar motility, as shown by extensive experiments [28–31]. Based on the current knowledge of the chemosensory system of *Campylobacterota*, we attempted to further approach the central question of "how and why diversity evolves" by tracking the evolutionary process of both the chemosensory system and flagellar machinery. We inferred the evolutionary events for gene gain, loss, and innovation that resulted in the diversity and complexity of the chemosensory system, and our results suggest that flagellar alteration can lead to F class switch or formation

of F class hybrid. Unexpectedly, our analyses also revealed that the complex flagellar motor structure with additional scaffolds evolved in the last common ancestor of *Campylobacterota*, whereas simpler structures in lineages such as the *Arcobacter* genus appeared later due to gene rearrangement and gene loss. These observations suggest an alternative hypothesis to the current understanding that the complexity of the flagellar motor incrementally evolved from simplicity [32].

## Results and discussions

### The conservation of F3 class in *Campylobacterota* is due to its primary role in flagellar motility

Our previous analyses revealed that the F3 class is conserved throughout the *Campylobacterota* phylum from hydrothermal vent specialists to host-associated species, except for unflagellated species and the two genera *Nitratiruptor* and *Arcobacter*, which will be discussed later in this paper [22]. We questioned why the F3 class is highly conserved even though other F classes undergo dynamic gain and loss in this phylum.

The F3 class is the only chemosensory class in the ancestral lineage of *Campylobacterota* at the deep-sea hydrothermal vent, including four genera, namely *Nautilia*, *Cetia*, *Caminibacter*, *Lebetimonas* that are all flagellated and motile (Fig 1A and S2 and S3 Tables). Thus, similar to the host-associated *H. pylori* and *C. jejuni*, the F3 class as the sole chemosensory class in these species is most likely controlling the flagellar motility. Although there is no experimental data for chemosensory system in *Campylobacterota* species with multiple F classes, we try to search for genomic clues to infer which F class is most important for flagellar motility. It is known that there is a significant correlation between chemosensory classes and chemoreceptor types, the latter of which refer to the number of helical heptads in the conserved signaling domain of chemoreceptors [23, 33]. For example, the F7 class has specific 36H chemoreceptors, while the F8 class with 34H and the F9 class with 44H [2]. For species with extra F classes in addition to F3 class, such as genera *Sulfurimonas*, *Sulfuricurvum* and *Sulfurospirillum*, we notice that they mainly have 40H chemoreceptors that are the input types of the F3 class (Fig 1B and S4 Table). In contrast, other F classes in F3-containing species generally have one or two cognate chemoreceptors encoded in the same gene cluster (Fig 1A and 1B), showing limited sensory ability and thus are less likely to serve as the major navigator for flagellar motility.

The above analyses show that all F3-containing *Campylobacterota* species have F3 cognate chemoreceptor types (40H and 28H) as the dominant signal input (Fig 1B). Based on this fact, it is tempting to infer that F3 class coupled flagella as its direct signaling output from the ancestor of this phylum and remained as the major chemosensory array in later lineages with multiple F classes. In support of this notion, all F3-containing species have the same set of flagellar genes, which will be discussed later. In addition, two lineages of *Campylobacterota* (*Sulfurovum*/*Nitratifractor* genera and a subclade of the *Campylobacter* genus) do not have flagella and have lost all chemosensory genes, suggesting the co-evolution of the chemosensory system and flagellar motility (Fig 1A and S2 Table).

The F3 class is exclusively present in *Campylobacterota* species [2, 34] and we next wonder why this class is unique to this phylum. F3 class was initially defined by only three phylogenomic markers: a core gene operon *cheVAW*, CheA protein with one additional receiver domain, and an auxiliary CheB lacking the receiver domain [2]. Other components that make a full chemosensory pathway, such as *cheY* and its phosphatase genes, are not in close proximity to the core gene operon; thus it is difficult to distinguish their F class belongings in species with multiple F classes. Since CheY and CheZ homologs are experimentally confirmed in *H. pylori* and *C. jejuni* that only contain one F3 class [35–37], these two proteins are considered as

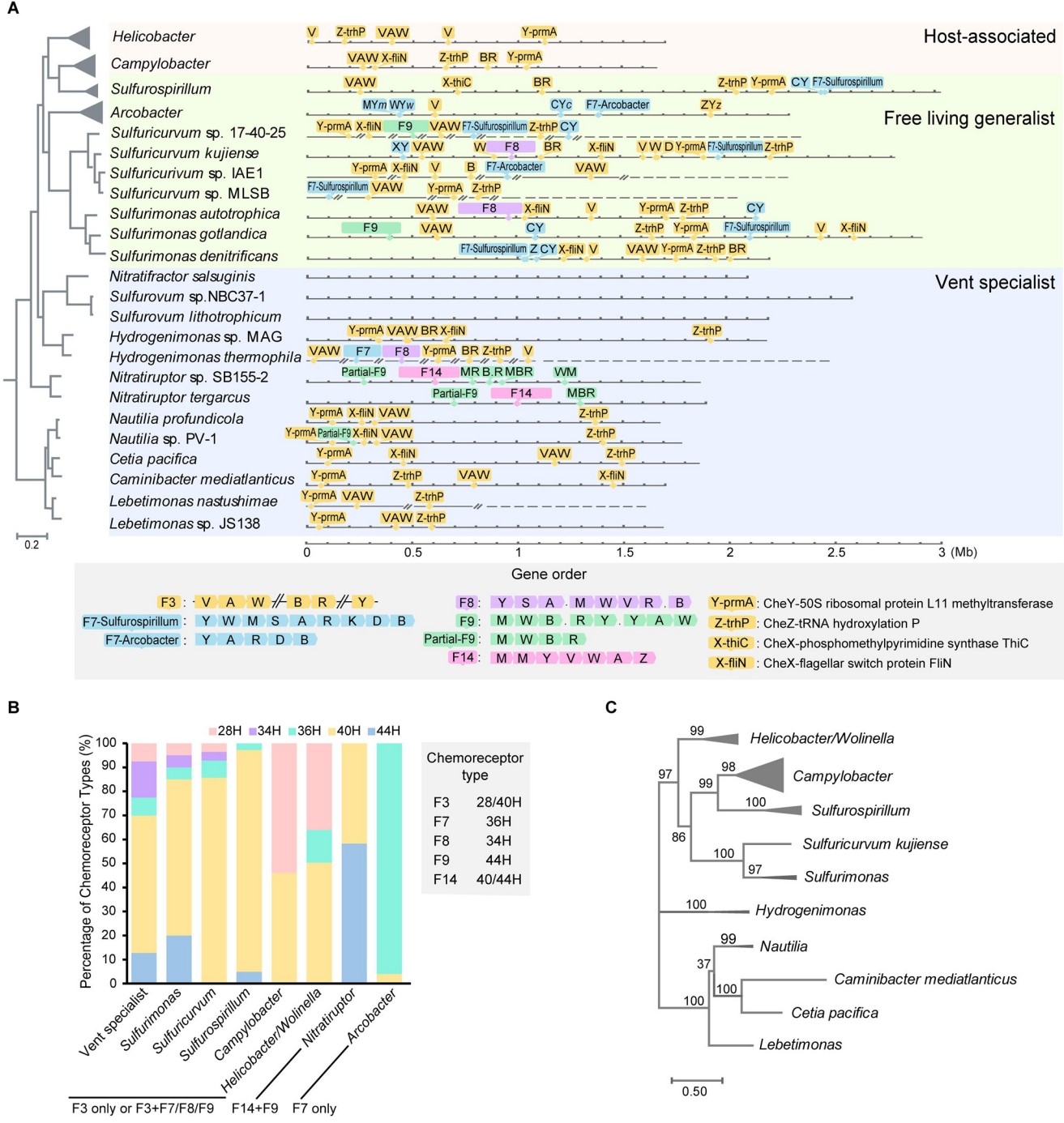

**Fig 1. Chemosensory system in *Campylobacterota*.** (A) The presence of chemosensory classes in representative species of *Campylobacterota* that are illustrated in linearized genomes and mapped to the species tree. The characteristics of F3/F7/F8/F9/F14 classes in terms of composition and gene order are depicted in the bottom box. Abbreviations for chemosensory genes: A, *cheA*; V, *cheV*; W, *cheW*; B, *cheB*; R, *cheR*; C, *cheC*; D, *cheD*; X, *cheX*; Y, *cheY*; Z, *cheZ*; M, chemoreceptor. "." represents a gene that is not involved in chemosensory system or of unknown function. The background colors are used to discriminate vent specialists, generalists and host-associated species of *Campylobacterota*. (B) The helical heptad types of chemoreceptors in *Campylobacterota*. The stacking histogram shows the percentage of each chemoreceptor type in respective genera, and the F classes of these genera are marked below. The cognate chemoreceptors types of F3/F7/F8/F9/F14 classes are shown in the right box. (C) Phylogenetic tree based on F3 class components (CheY, CheZ and CheX) in *Campylobacterota*.

prototypes of F3-CheY and CheZ in our analyses. We noticed that in all *Campylobacterota* genomes with the F3 class, the downstream genes of *cheY* and *cheZ*, corresponding *prmA* and *trhP* (previously annotated as U32 peptidase), encode highly conserved proteins that are easily identified through homology search, and their upstream genes are consistently *cheY* and *cheZ* (Figs 1A and S1). In addition, we also identified that all F3-containing *Campylobacterota* genomes encode a possible CheX homolog with conserved CheX motif and distinct from CheC and CheZ (S2 and S3 Figs). The downstream genes of this potential *cheX* are mostly *fliNO* in *Campylobacterota* genomes with the F3 class, except that *thiC* is adjacent to *cheX* in the *Sulfurospirillum* genus and *cheX* is lost in some *Helicobacter* species (Figs 1A and S1 and S3 Table).

The conserved gene orders of *cheY-prmA*, *cheZ-trhP* and *cheX-fliNO* in the *Campylobacterota* phylum are unusual because they are not functionally related, and it is known that the gene order is poorly conserved in bacteria [38, 39]. Their strict co-occurrence with the F3 class, especially in both deep-branching and later host-associated species that only have one F3 class, strongly support their association with this class. In addition, the phylogenetic tree based on concatenated CheY, CheZ, and CheX encoded by the above gene clusters showed similar topology to the species tree, indicative of vertical inheritance, similar to core components CheV, CheA, and CheW of the F3 class (Fig 1C) [22]. Thus, the highly scattered genomic distribution but conserved downstream gene orders of *cheY/Z/X* are new features of the F3 class. Moreover, because of the dispersed distribution across the genomes, the F3 components cannot be transferred in a single event, thus explaining the specificity of the F3 class to the *Campylobacterota* phylum.

## F class switches results from entire flagellar alterations

Notably, the F3 class was missing in two *Campylobacterota* genera, namely *Nitratiruptor* and *Arcobacter*, which include species that are flagellated and motile (S2 Table). *Nitratiruptor* species have a complete F14 class with a partial gene set of the F9 class, whereas *Arcobacter* species only have the F7 class (Fig 1A and S3 Table). These F class switches also changed the signal input repertoire, as seen in the fact that these two genera have chemoreceptor types distinct from the rest of this phylum (Fig 1B). Chemoreceptors of *Arcobacter* species generally belong to 36H that are compatible with the F7 class, whereas half of the chemoreceptors of *Nitratiruptor* species belong to 44H, a cognate type of the F9 class (Fig 1B and S4 Table) [2]. If these non-F3 classes confer greater fitness advantage over F3, F class switches would occur frequently in species with rampant HGT events, and F3 would not be a conserved type throughout this phylum. To investigate this puzzle, we examined co-evolved signaling output-flagellar genes.

Remarkably, *Campylobacterota* species with the F3 class have extremely scattered flagellar gene distribution, with >50 genes in approximately 30 discrete loci throughout their genomes (Figs 2A and S4) [40]. In contrast, *Nitratiruptor* species have their F14 class and almost all flagellar genes in one large cluster, except for a few genes such as *motAB* and *flhX-fliK*. Phylogenetic analyses showed that both their chemosensory and flagellar components branch with *Aquificota* phylum, particularly *Persephonella* species (Fig 2B) [22]. In addition, *Persephonella* species have been isolated from deep-sea hydrothermal vents, which are the same habitats in which *Nitratiruptor* species were found. These data strongly suggest that *Nitratiruptor* acquired the F14 class and an entirely new flagellar gene set through HGT from an *Aquificota* species but lost its native components. Furthermore, we speculate a "loss first then gain" process rather than coexistence followed by selective loss based on two points of evidence: (i). The gene order of the transferred DNA is almost identical in both the donor and recipient, indicative of a recent transfer event (Figs 2C and S5); (ii). The original F3 chemosensory and flagellar

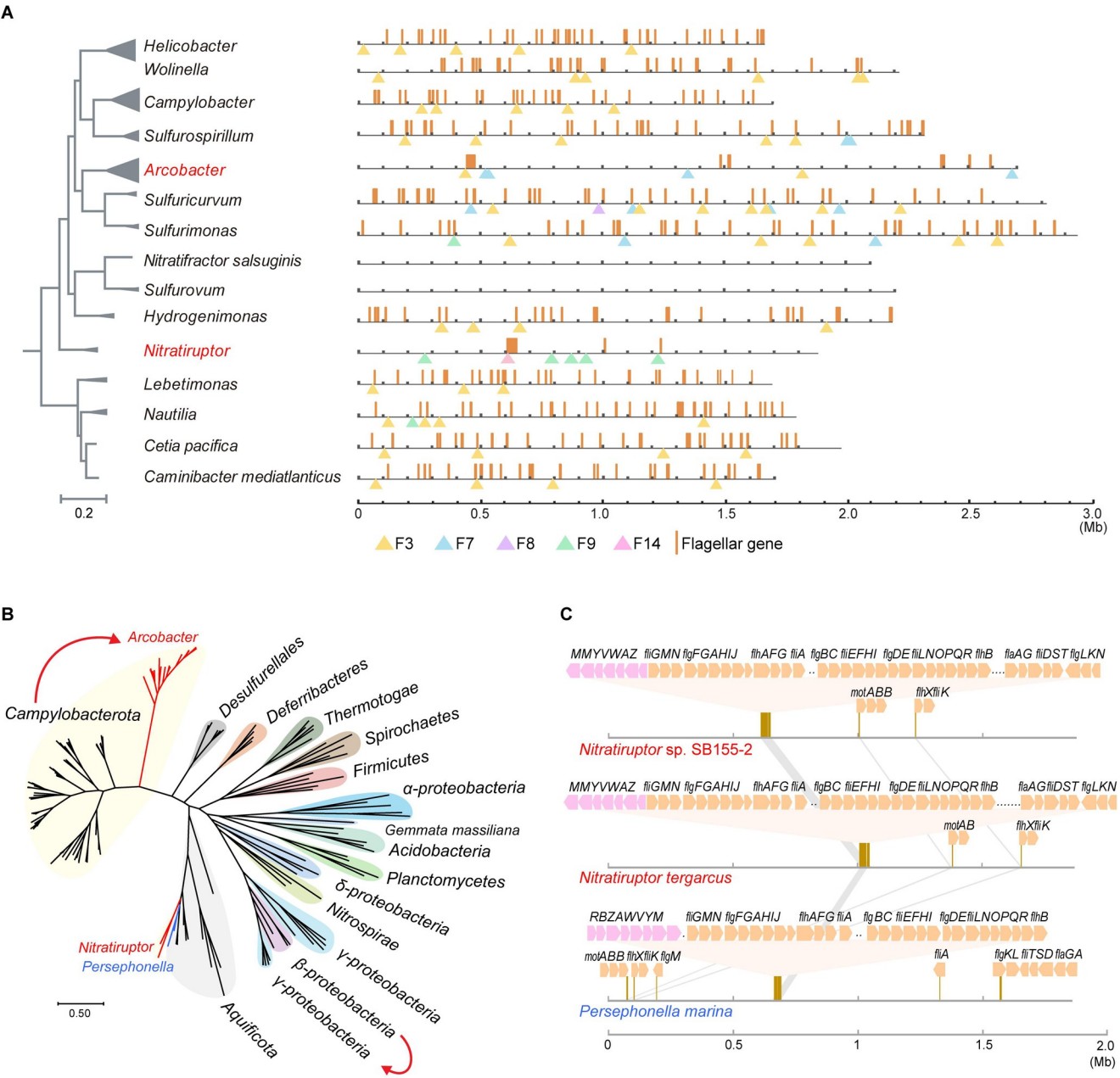

**Fig 2. F class switches and flagellar alterations.** (A) The distribution of flagellar genes are illustrated by brown lines in linearized genomes of representative species from *Campylobacterota*. The triangles below point out the location of a chemosensory gene locus on the chromosome. *Nitratiruptor* and *Arcobacter* are highlighted in red because their chemosensory class for flagellar motility are different from the rest genera. (B) Phylogenetic tree based on concatenated alignments of conserved flagellar proteins including FlhA, FlhB, FliF, FliG, FliE, FlgB, FlgC, FlgK and FlgL. The red curves highlight close evolutionary relationships of flagellar genes between two clusters, starting from the gene source with an arrow pointing to the recipient. (C) The gene order of F14 chemosensory class and flagellar locus in species of *Nitratiruptor* and *Persephonella*. The flagellar gene loci on the linearized genomes are zoomed in to show gene orders of flagellar (orange arrow) and chemosensory genes (pink arrow).

genes were completely removed from the genome without mixing them with the newly acquired gene set (Fig 2A and 2C).

*Arcobacter* species have most of their flagellar genes at two main loci, with approximately 30 genes at one locus and approximately 10 at the other (Figs 2A and 3A and S6). Phylogenetic

analyses based on concatenated flagellar proteins showed that *Arcobacter* spp. formed a cluster peripheral to the F3 class-containing *Campylobacterota* group, suggesting a common origin, but incongruent with the phylogenetic position of the species (Fig 2B). Compared to species from an ancestral genus (*Lebetimonas*) and a closely related genus (*Sulfurimonas*), the flagellar gene orders of *Arcobacter* were shuffled across the genome (Figs 3A and S6). There were no known intermediate clustering levels between the *Arcobacter* genus and the rest of *Campylobacterota* (Figs 2A and S4). In addition, the two large flagellar loci of *Arcobacter* genomes shown in Fig 3A did not simply piece together smaller ancestral clusters, since ancestral clusters conserved in all F3 class-containing *Campylobacterota* genomes, such as *flhFG-flgV-fliAMY*, *flgIJMNK*, *fliK-flgDE* and *flgBC-fliE*, were broken and shuffled in *Arcobacter* genomes (Figs 3A and S7). It is well known that the expression of flagellar genes is tightly regulated in a hierarchical manner to coordinate the highly ordered flagellar assembly process [41–44]. However, orderly expressed Class 1, 2 and 3 genes in both *C. jejuni* and *H. pylori* were mixed in the two large flagellar loci of *Arcobacter*, suggesting that cluster formation was not driven by transcriptional regulation (Figs 3A and S7). Furthermore, flagellar-specific regulators in both *C. jejuni* and *H. pylori*, such as RpoN, FlgRS, and FliA (also its antagonist FlgM), were conserved in all F3 class-containing *Campylobacterota* species but were lost in all *Arcobacter* species. Additionally, no genes encoding DNA-binding regulators were found within or near the flagellar loci (Figs 3A and S7). Thus, the flagellar structural genes in *Arcobacter* species share the same origin as other *Campylobacterota* species, except for the *Nitratiruptor* genus, but are regulated by different and yet unknown regulators.

Collectively, the F class switch in *Nitratiruptor* is due to HGT of entirely new chemosensory and flagellar gene sets together from another bacterial phylum. In the ancestor of *Arcobacter* species, the ancestral F3 class is lost, accompanied by massive genetic rearrangements of flagellar structural genes and the loss of ancestral flagellar regulator genes, whereas the laterally

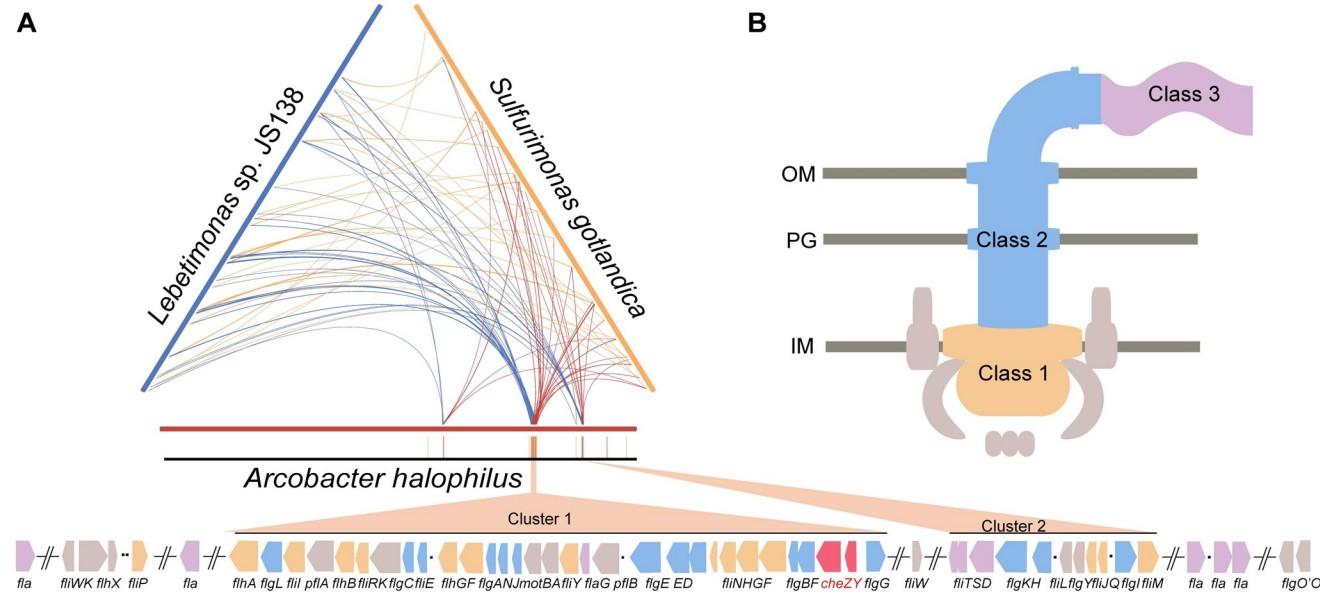

**Fig 3. Flagellar genes in *Arcobacter*.** (A) Ternary comparison of flagellar gene distribution in genomes of *Arcobacter halophilus* (red), *Lebetimonas* sp. JS138 (navy) and *Sulfurimonas gotlandica* (yellow). The same flagellar genes are linked with each two of the three chromosomes. Ochre strips below represent the flagellar genes on the linearized genome of *Arcobacter halophilus* beneath the ternary pattern, and the gene orders of two main flagellar loci (Cluster 1 and Cluster 2) are shown at the bottom. (B) The regulatory hierarchy of flagellar genes in *C. jejuni* and *H. pylori*. The colors represent the transcriptional programing of flagellar genes starting from Class 1, then Class 2, to Class 3. The gene clusters in Fig 3A at the bottom are shaded based on the same color scheme, except two chemosensory genes *cheZY* in red.

acquired F7 class vertically passes on to all its decedents with gene expansion of 36H chemore-ceptors. Thus, the F class switches investigated here result from alterations in the entire flagel-lar genes, whereas the F3 class is constantly preserved in species with no substantial changes in the original flagellar genes, supporting its role in controlling flagellar motility. Importantly, the evolutionary episode of flagellar genes in the *Arcobacter* genus cannot be explained by mobile element-mediated relocation or HGT in a stepwise manner but deserves in-depth exploration, which may uncover unknown mechanisms of genome stability and evolution.

## Rare hybrids of cytoplasmic F9 class with other F class(es)

The F class components of the above two genera that experienced revolutionary changes in chemosensory and flagellar genes, are not of the same origin. The most distinctive case is *Nitratiruptor* with a full F14 class and a partial gene set of the F9 class. Components of F9 class have unusual domain architectures unique to this class: chemoreceptor DosM has double sig-naling domains in tandem and adaptor CheW contains concatenated triple CheW domains (hereafter called $CheW_3$) [2, 17]. As shown in Fig 1A, two *Nitratiruptor* species have F9 genes, namely *dosM*, $cheW_3$ and *cheBR*, but lack kinase *cheA* and response regulator *cheY* in the same gene cluster. In addition, no other *cheA* genes were found in these two *Nitratiruptor* genomes, except for one copy in the F14 cluster (S3 Table). The same partial gene set of the F9 cluster was found in one *Nautilia* species with a full F3 class that belongs to the ancestral lineage dwelling in deep-sea hydrothermal vents (Fig 1A). To assess the abundance of the partial F9 class in the *Bacteria* domain, 2781 representative species with complete genomes across diverse bacterial phyla were examined for the presence of both *dosM* and $cheW_3$. Only eight species including the three *Campylobacterota* species described above have a partial F9 gene set, sug-gesting that a partial F9 gene set is rare in bacteria (Fig 4A and S5 and S6 Tables). In addition, full F9 class gene clusters were only found in 64 out of 2781 species (approximately 2%) with very sporadic species distribution (Fig 4A and S5 and S6 Tables). Phylogenetic analyses based on DosM and $CheW_3$ suggested that F9 class gene clusters underwent many HGT events, because few bacterial phyla formed a monophyletic cluster in the tree, in addition to its spo-radic phylogenetic profile (Fig 4B and 4A). Moreover, the partial F9 gene set evolved indepen-dently at least three times in *Campylobacterota*/*Aquificota*, *Firmicutes*, and α-proteobacteria, and later underwent HGT and gene duplication events (Fig 4B).

Interestingly, the F9 class is rarely present alone in species (only 3 out of 72 representative species have the F9 class alone), and is more often encoded in genomes that also have other F class(es) (Fig 4B). Among the species with the F9 class, 97% had a complete set of flagellar genes (Fig 4B). A recent electron cryotomography study of *V. cholerae* with a full F9 class revealed that F9 components form two-layered cytoplasmic arrays that are flat and stabilized by DosM [10, 17]. It is possible that the purely cytoplasmic location and flat structure of the F9 array prevent it from monitoring extracellular environments; thus, bacterial species with the F9 class tend to employ another F class to engage more transmembrane chemoreceptors. In addition, F9 arrays were observed in *V. cholerae* under carbon starvation or low-oxygen condi-tions, but not during the exponential phase in rich medium, suggesting that F9 array assembly is conditionally induced and mostly related to stress [17, 45]. Collectively, the F9 class generally co-exists with other F class(es) in bacterial species and is less likely to serve as the primary nav-igator for flagellar motility, although its physiological effects remain to be discovered.

Notably, 98% of the DosM homologs identified here contained only two signaling domains acting as structural scaffolds but lacked sensory domains, indicating that the double-layered arrays need to incorporate additional chemoreceptors or sensory proteins for signal input (S6 Table). Consistently, electron cryotomography of F9 arrays in *V. cholerae* showed DosM in

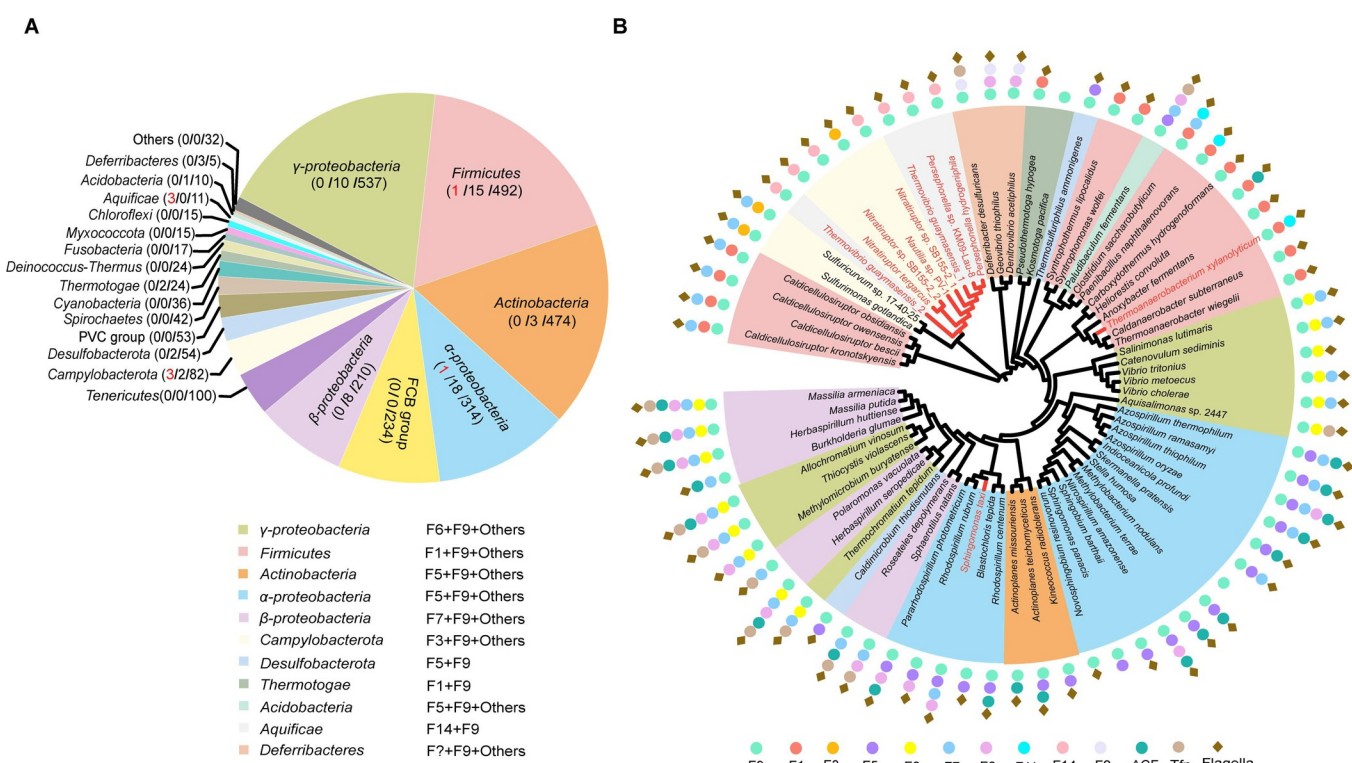

**Fig 4. F9 class in the *Bacteria* domain.** (A) Distribution of F9 class in different bacterial taxon. Numbers in parentheses of the pie chart represent the number of the species containing partial F9 class / the number of the species containing full F9 class/ the number of species investigated in the taxon. (B) Phylogenetic tree based on concatenated DosM and CheW$_3$. The presence of different chemosensory classes and flagellar genes in these species are depicted around the tree. The species that have partial F9 gene set are highlighted in red.

trimer formation with other cytoplasmic chemoreceptors at both baseplates [17]. For species with a partial F9 gene set, the cytoplasmic arrays can still form with DosM and CheW$_3$ since the same study in *V. cholerae* revealed that F9-CheA is not essential for array formation [17]. However, to achieve signal transduction, the partial F9 class possibly cooperate with other F class components particularly CheA and CheY. For example, DosM and CheW$_3$ likely interact with F14-CheA in *Nitratiruptor* species and F3-CheA in *Nautilia* sp. PV-1, since their genomes only encode one copy of *cheA* (Fig 1A and S3 Table). However, partial F9 class is rare in bacteria (only eight representative species) and our sequence analyses did not yield potential amino acid variation in DosM/CheW$_3$ or CheA of other F classes that specifically mediates the interaction of the F class hybrids.

## F class hybrids concomitant with flagellar alterations

Another F class hybrid was found in *Arcobacter*. As shown in Fig 1A, although the F7 class operon (*cheYARDB*) in *Arcobacter* genomes encodes the response regulator gene *cheY*, there are four additional copies of *cheY* dispersed in the genomes. These genes were annotated as *cheY* here because each of them is adjacent to another chemosensory gene, such as a chemoreceptor gene, *cheW*, *cheC* or *cheZ*, suggesting that they most likely function in chemosensory rather than in two-component system. To distinguish these genes, they are named after their adjacent chemosensory genes as CheY*m*, CheY*w*, CheY*c*, and CheY*z*, in addition to F7-CheY in the F7 operon (Fig 1A and S7 Table). Phylogenetic tree analysis of all CheY homologs in *Campylobacterota* showed that all F7-CheY homologs formed a cluster, whereas CheY*z*

proteins branched with F3-CheYs (Fig 5A). Since CheY*z* and its phosphatase CheZ are encoded in the largest flagellar locus in *Arcobacter* genomes, it is likely that CheY*z* came from the ancestral F3 class and shuffled together with flagellar genes in the last common ancestor of *Arcobacter* species (Fig 3A). In addition, CheY*m*, CheY*w*, and CheY*c* formed independent clusters, and all three clusters branched together in the phylogenetic tree, suggesting that they may have originated from gene duplication events, although their F class origin is unclear (Fig 5A).

The fact that the F7 gene cluster is the only F class with the most transmitter genes and its cognate 36H chemoreceptors are the dominant type in *Arcobacter* genomes suggests that F7 components likely control flagellar motility in *Arcobacter* species (Fig 1A and 1B). Thus, an interesting question arises regarding which one of the five CheY homologs directly interacts with flagellar motor switch proteins such as FliM. Our analyses suggest that F3-CheY controls flagellar motility in F3 class-containing species, and F14-CheY, as the only CheY in

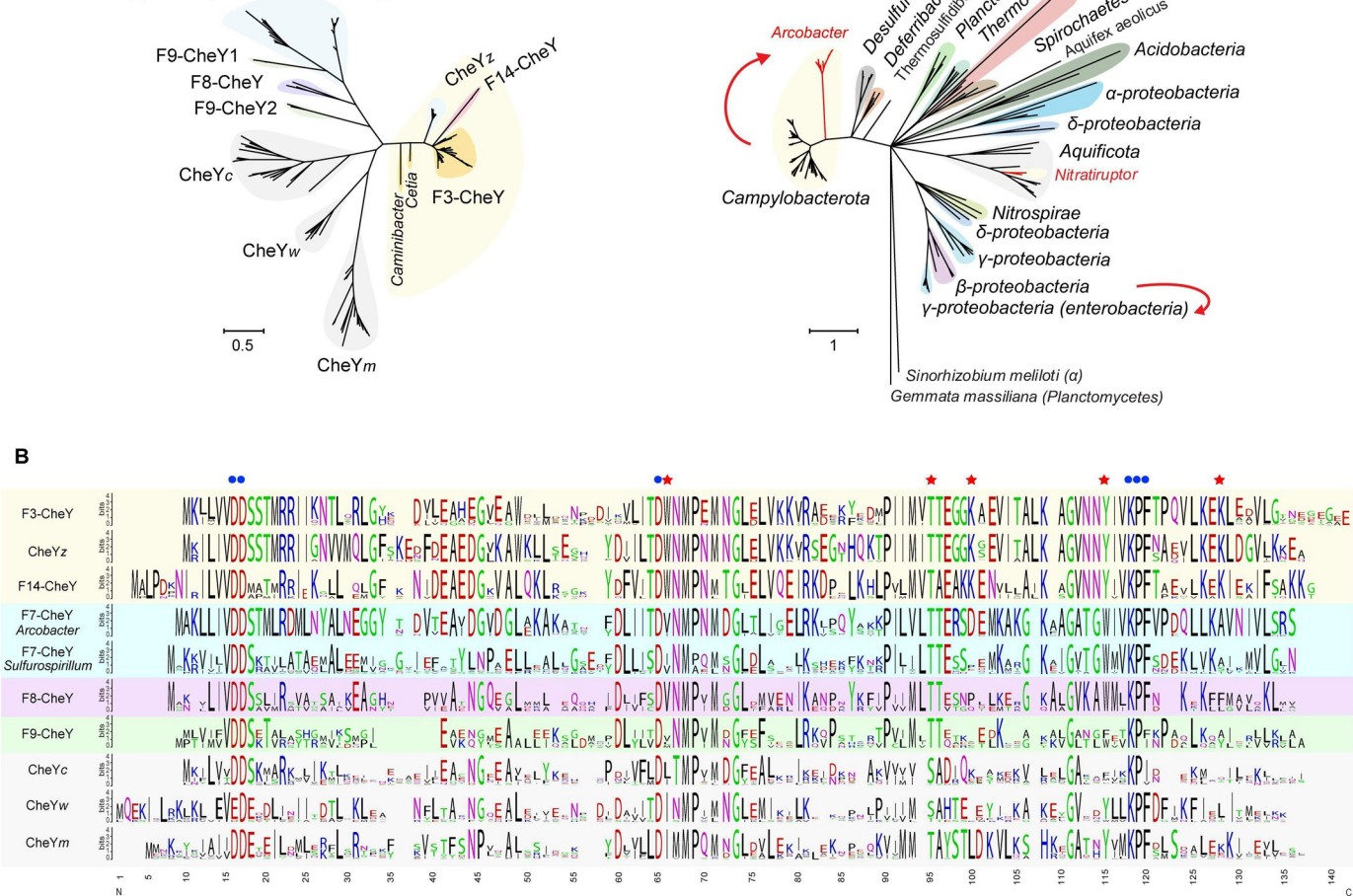

**Fig 5. F class hybrids in *Arcobacter*.** (A) Phylogenetic tree of all CheY homologs in *Campylobacterota*. CheY homologs that likely interact with the flagellar motor switch proteins are shaded by light yellow. (B) Sequence alignments of all CheY homologs in *Campylobacterota*, represented by sequence logos for each CheY cluster based on the phylogenetic tree of Fig 5A. The blue dots mark conserved residues for almost all CheY homologs, and the red stars tag the key residues for binding FliM. (C) Phylogenetic tree based on FliM protein from different bacterial phyla. The red curves highlight close evolutionary relationships of FliM homologs between two clusters, starting from the gene source with an arrow pointing to the recipient.

*Nitratiruptor* species, must interact with its co-transferred flagellar components. Because CheY$z$ forms a distinctive cluster with F3-CheY and F14-CheY in the phylogenetic tree, it is possible that CheY$z$ also interacts with the flagellar motor in *Arcobacter* (Fig 5A). Previous structural and mutagenesis studies of CheY homologs from *E. coli* and *H. pylori* have revealed key residues for binding FliM, including $W^{58}$, $K^{91}$, $K^{119}$, and coupling $Y^{106}$-$T^{84}$ (residue numbers corresponding to *E. coli* CheY) [12, 13, 37]. Three residues $W^{58}$, $K^{91}$, and $K^{119}$, are exclusively conserved in F3-CheY, F14-CheY, and CheY$z$. Coupling $Y^{106}$-$T^{84}$ are conserved in F3-CheY, F14-CheY, CheY$z$, and CheY$m$, but not in the rest of the CheY homologs from *Campylobacterota* species (Fig 5B). These results suggest that CheY$z$, rather than F7-CheY, is the response regulator for the interaction with FliM in *Arcobacter*, even though its kinase CheA and input signal receptors belong to the F7 class.

The branching pattern of CheY$z$ within the F3-CheY cluster and its gene location inside a flagellar locus that shares the same origin as flagellar genes from F3 class-containing species indicate that *cheYZ* are remnant genes from the ancestral F3 class but diverged from the rest of F3-CheY. Phylogenetic analysis of its interaction partner FliM revealed the same branching pattern regarding relationships of homologs from *Arcobacter* and F3 class-containing *Campylobacterota* species, suggesting co-evolution of CheY and FliM, presumably due to protein-protein interactions (Fig 5A and 5C). However, co-evolution is not limited to these two interacting partners, because, as described above, the phylogenetic tree based on concatenated flagellar proteins showed the same clustering pattern as the trees based on FliM or CheY (Figs 2B and 5A and 5C). These results suggest that F3-*cheYZ* followed the same evolutionary path as flagellar genes in the ancestors of *Arcobacter*, and the laterally acquired F7 class likely takes control of flagella through incorporation of F3-*cheYZ*. Furthermore, the concomitant events of all flagellar alterations and the F class switch/hybrid are not unique to *Arcobacter*. A similar F class hybrid was recently reported for *E. coli* by tracking the evolution of F6 and F7 classes in β- and γ-proteobacteria, but whole flagellar changes in enteric γ-proteobacteria have been described by another study without linking these two concurrent evolutionary events [21, 46].

## Complex flagellar motor evolved in the ancestor of *Campylobacterota*

Our detailed analyses of the two F class switches uncovered two independent revolutionary changes in flagellar genes in the two genera of *Campylobacterota*. As the latest lineages of this phylum, the human pathogens *C. jejuni* and *H. pylori* have been shown to possess the most complicated flagellar motor structures that can generate high torque to enable swimming at high speeds through viscous media [47–50]. Thus, we investigated how this complex structure has evolved in the *Campylobacterota* phylum.

The genomes of 82 representative species of *Campylobacterota* were examined for all known flagellar genes that encode structural components, chaperones, accessory proteins, and regulators. In addition to the 36 core flagellar proteins commonly present in flagellated bacterial species, we summarized 20 proteins that were either specific to this phylum or combined with other components showing structural features distinctive from other bacterial phyla (S8 Table). The presence or absence of these 20 components was mapped to a robust species tree of the *Campylobacterota* phylum (Fig 6A). Surprisingly, we found that all F3 chemosensory class-containing *Campylobacterota* species, including the ancestral lineage from deep-sea hydrothermal vents, had the same flagellar composition as *C. jejuni* and *H. pylori*. In particular, they share proteins that constitute unique torque-generating motor structures of the two pathogens, including the following:

i. FlgP, FlgQ, PflA and PflB form three prominent disk complexes (basal, medial, and proximal disks) that are essential for scaffolding an outstanding number of stator units (17 in *C. jejuni* and 18 in *H. pylori*, compared to approximately 11 stator units in *E. coli*) [48, 49].

ii. FliY, which is mutually exclusive with FliN in most bacterial species, but co-occurs with FliN in this phylum, together with FliM, produces a wider C-ring to scale with the expanded stator ring [51].

iii. FlgX, a chaperone for the stator units, protects their integrity (Fig 6A) [52].

These proteins that are not found in the model organism *E. coli* serve as structural scaffolds (FlgPQ, PflAB), torque-switching rotors (FliY with FliMN), or chaperones (FlgX) for the incorporation of extra stator units, which are required for high-performance motility in *C. jejuni* and *H. pylori*. Thus, their presence in all F3 class-containing *Campylobacterota* species suggests that these species likely have basal, medial, and proximal disks in combination with a wider C-ring to support more stator units, similar to that in *C. jejuni* and *H. pylori.*

The structural complexity and high-torque generation of *C. jejuni* and *H. pylori* may require additional components that have recently been functionally characterized, in addition to the above proteins. Notably, PflB and FlgX were first identified as flagellar components from a genome-wide screening of *C. jejuni* which also uncovered several other new flagellar proteins that were not found in the *E.coli* model, including FlgV, FlgW, and FlgY [53]. Although their exact roles or position in the flagellar apparatus remain unexplored, they may be candidates for the assembly or function of complex motors. Thus, homologs of FlgV, FlgW, and FlgY were examined in all representative species of *Campylobacterota*, and they were present in all species with the F3 class, even with highly conserved gene orders in the genomes (Figs 6A and S8). In addition, flagellar specific regulators such as RpoN, FlgRS, FliA-FlgM, and CsrA-FliW were also investigated, and all of them were found in *Campylobacterota* species with the F3 class (Fig 6A).

Collectively, our genomic analyses of flagellar compositions in *Campylobacterota* suggest that the flagellar motor in the last common ancestor of this phylum is a complex structure with additional scaffolds (Fig 6B). In addition, it is likely that the ancestor also evolved a complicated regulatory hierarchy to control the assembly of flagellar nanomachines, because the same set of regulator genes are conserved from species in deep-sea hydrothermal vents to host-associated pathogens. Although most lineages of this phylum preserved complex flagellar structures and regulatory programs through vertical inheritance, some lineages did not follow. *Sulfurovum* was the dominant group in the deep-sea vent biofilm and its close relative *Nitratifractor* lost all flagellar and chemosensory genes (Figs 6A and 1A and 2A). The same was observed in a small clade within the *Campylobacter* genus (represented by *C. gracilis*, *C. hominis*, and *C. ureolyticus*), perhaps because of lifestyle changes associated with biofilms or hosts (S4 Fig and S2 Table). As described, *Nitratiruptor* lost the original flagellar genes and F3 class but gained another flagellar gene set and F14 class from *Aquificota* through HGT. Compared with the original flagellar composition, the gene cluster from *Aquificota* lacked genes encoding FlgPQ (basal disk), FliY (wider C-ring), FlgX (stator chaperon), RpoN/FlgRS/FlgM/CsrA/FliW (regulatory checkpoints) and FlgVW (unknown function) (Fig 6A and S8 Table).

The last and most intriguing case is *Arcobacter*, with flagellar genes that mostly share the same origin as homologs from F3 class-containing *Campylobacterota* species, but their genomic location has been extensively shuffled and their sequences have diverged from the other *Campylobacterota* species. Moreover, genes encoding FlgPQ and FlgVWX that either constitute complex motor scaffolds or are specific to *Campylobacterota* were not found in *Arcobacter*

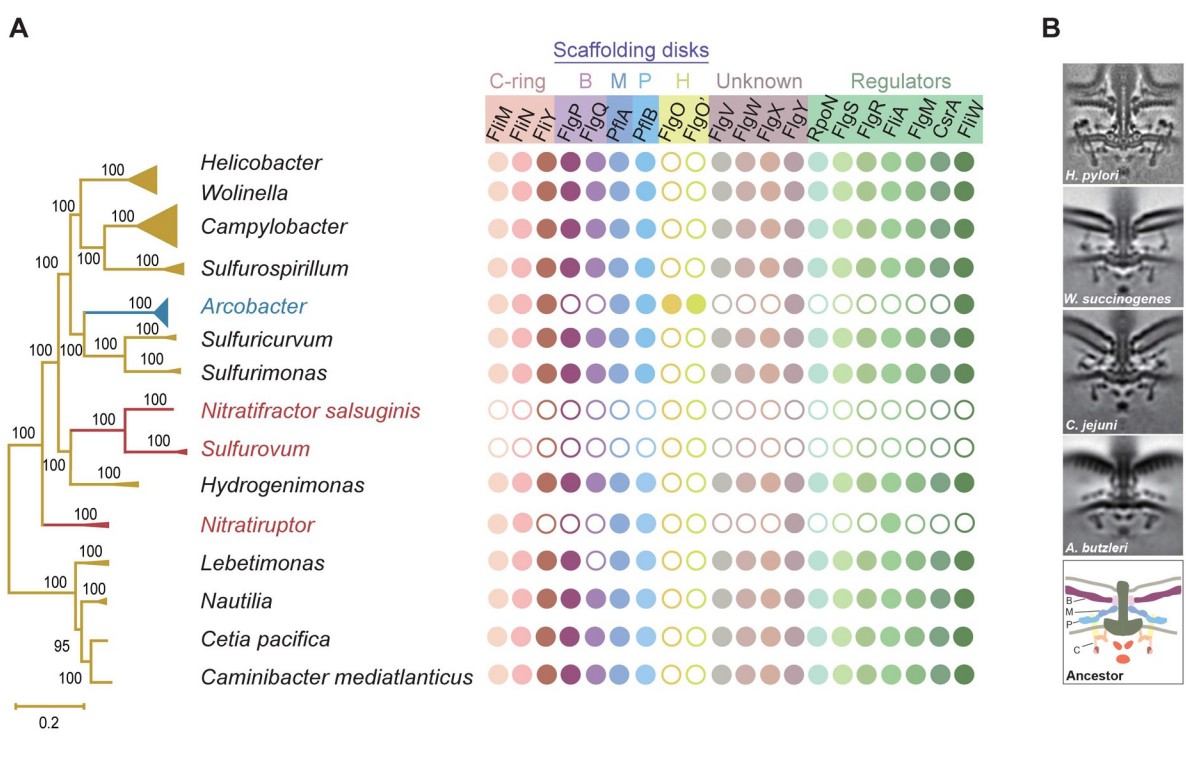

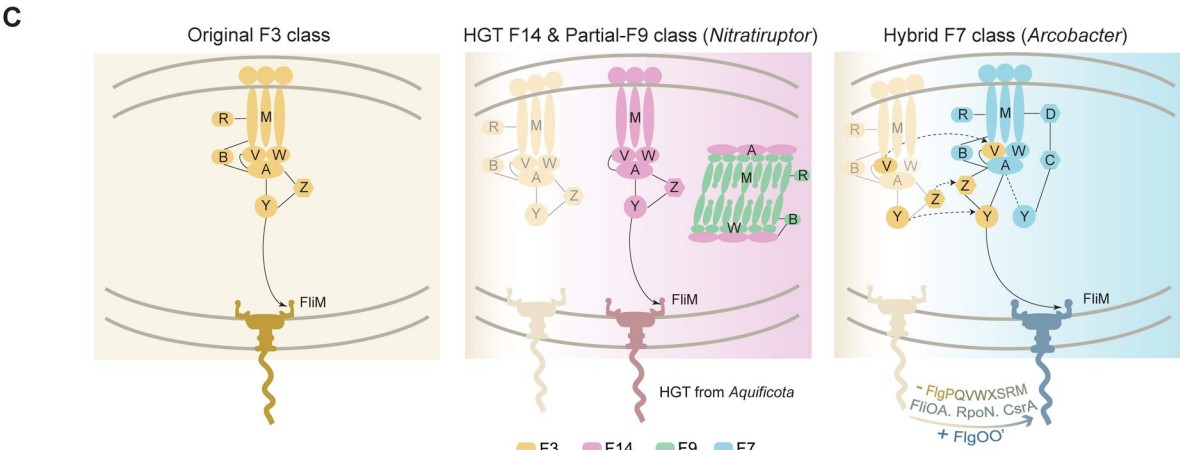

**Fig 6. Co-evolution of flagellar motor and chemosensory system in *Campylobacterota*.** (A) The presence or absence of 20 flagellar components mapped to the phylogenetic tree of *Campylobacterota*. Dot represents the presence of the specific gene and circle means absence. B, Basal disk; M, Medial disk; P, Proximal disk; H: H-ring. (B) The electron cryotomography pictures of flagellar motors in representative species including *H. pylori* [50], *Wolinella succinogenes* [32], *C. jejuni* [48], *Arcobacter butzleri* [32], and a proposed flagellar model for the ancestor of *Campylobacterota*. (C) Working model for the evolution of chemosensory system and flagella in *Campylobacterota*.

genomes (Fig 6A). It is likely that these non-core flagellar genes were lost during flagellar alterations in the ancestor of *Arcobacter*, similar to the gene loss of the regulators RpoN, FlgRS, FliA-FlgM, and CsrA (Fig 6A). However, *Arcobacter* acquired an additional flagellar protein, FlgO, and a potential FlgO paralog, which show a certain degree of similarity to each other and their encoding genes likely form an operon in *Arcobacter* genomes (Figs 6A and S9 and S10). A recent electron cryotomography study revealed that the *Arcobacter*-type motor is different from the *Campylobacter*-type motor regarding the lack of a basal disk made of FlgPQ, but it

possesses 5–6 concentric rings associated with the outer membrane (Fig 6B) [32]. These concentric rings are likely to be composed of FlgO, which forms similar rings (H-ring) in *Vibrio* species and is not found in other *Campylobacterota* species, except *Arcobacter* [54]. Additionally, the *Arcobacter*-type motor has only 16 stator units, which is less than *C. jejuni* (17 units) and *H. pylori* (18 units) [32, 49]. Overall, both compositional analyses and structural observations suggest that the *Arcobacter*-type motor is simpler than the motor type of F3 class-containing *Campylobacterota* species. However, the simplicity in structure and peripheral phylogenetic position with other *Campylobacterota* species in trees based on flagellar proteins does not mean that the *Arcobacter*-type motor is the ancestral form of this phylum. Our analyses clearly suggest that the ancestral lineage of *Campylobacterota* from deep-sea hydrothermal vents already has complex flagellar compositions similar to that of *C. jejuni* and *H. pylori*, which emerged recently (Fig 6A and 6B).

## Conclusions and outlook

The concept of co-evolution of chemotaxis and flagellar motility has long been well accepted based on their co-occurrence in species; however, detailed molecular evidence for their joined evolutionary changes is lacking. Here, using the *Campylobacterota* phylum as a case study, we clearly show that the chemosensory class responsible for flagellar motility will switch class or form a new hybrid class if flagellar genes undergo revolutionary changes (Fig 6C). Ecological adaptation may be the driver of diversification in both systems. In contrast, the primary chemosensory class for flagellar motility remain conserved if the flagellar gene set remains unaltered (Fig 6C). However, flagellar motility is only one of the signaling outputs for species with multiple chemosensory classes [15, 34]. The diversity and complexity of the chemosensory system are likely due to evolutionary changes in all signaling outputs, which requires further studies of chemosensory classes that do not target flagellar motility. Although these observations were only derived from the *Campylobacterota* phylum, these principles may apply to the entire *Bacteria* domain. More cases of co-evolution of the chemosensory system and flagellar gene set require comprehensive examination of all bacterial lineages that are flagellated. However, a robust branching order or relationships of bacterial lineages that are the basis for evolutionary history reconstruction remains notoriously difficult to resolve for the *Bacteria* domain [55].

Particularly, the F class switch with entire flagellar alterations in *Arcobacter* brings up an evolutionary puzzle: how and why approximately 50 building blocks of flagellar nanomachines shuffled across the genome? It cannot be explained by HGT as the case of *Nitratiruptor* for the following reasons: (i). *Arcobacter* spp. form an independent cluster in the phylogenetic tree based on flagellar proteins, peripheral to F3-containing *Campylobacterota* species but not closely related to any species or lineage that could be a specific HGT source; (ii). If the HGT source is an ancient F3-containing *Campylobacterota* species, how or how many events can achieve transfers of approximately 50 highly dispersed genes that are irreducible for both assembly and proper function? (iii) The flagellar loci in modern *Arcobacter* species are not fusion of conserved small flagellar gene clusters in F3-containing *Campylobacterota* species and their gene order does not reflect co-regulation of genes according to the assembly order, so what determines current flagellar gene organization? Answers to these require in-depth experimental studies of flagellar regulation, assembly and function in *Arcobacter* species that unfortunately lack a genetically tractable system now. In addition, comparative analyses of *Arcobacter* with evolutionarily more closely related *Campylobacterota* species may provide clue for the intermediate steps of these changes, but the close relatives of *Arcobacter* have not been identified yet or become extinct. This evolutionary puzzle is elaborated here not only

because of its significance in understanding flagellar evolution, but also in bacterial genome organization and evolution.

Beyond the *Campylobacterota* phylum, the F9 class was systematically investigated in species covering different phyla due to its unique structural features and application potential in synthetic biology. Recent electron cryotomography studies showed that the array structure made of the F9 class is purely cytoplasmic without membrane support, and as a result, it can be easily reconstructed *in vitro* [10]. Our comparative genomic analyses reinforced the idea that this double-layered array can incorporate other unknown chemoreceptors or sensory proteins for signal input, and cooperate with CheA and CheY from other chemosensory classes for signal output. Future studies on the assembly and signaling mechanisms of F9 arrays can facilitate its application as cell-free biosensors or signaling apparatus.

Finally, how complex flagellar motors such as those of *C. jejuni* and *H. pylori* evolved remains an interesting question [32, 56]. Our analyses within the *Campylobacterota* phylum suggest that complex motors likely evolved in the common ancestor of this phylum. Later lineages with simpler motor structures, such as *Arcobacter*, lost some structural components and their motors were not the primitive form for this phylum. To understand the origin and evolution of the high-torque flagellar motor, our analyses need to be extended from the *Campylobacterota* phylum. Other bacterial phyla also have complex but dissimilar flagellar motor structures. For example, the endoflagella of *Spirochaetes* species can produce torque of up to 4000 pN nm, compared to approximately 3600 pN nm of the *H. pylori* motor and approximately 1200 pN nm of the *E.coli* motor [57]. Recent electron cryotomography studies revealed that *Spirochaetes* motors have 16 stator units and a wider C-ring, but their scaffolding structures and compositional proteins are different from those of *Campylobacterota* [58–60]. Therefore, understanding the evolutionary paths to complex motor structures and high-torque generation requires a thorough investigation of the flagellar motors in the *Bacteria* domain rather than sampling limited model organisms. Importantly, the evolution of macromolecular complexes cannot be inferred based on only the phylogenetic tree of their compositional protein sequences but requires an organismal tree that can be rooted. Therefore, future electron cryotomography exploration of flagellar motor diversity and improved bacterial phylogeny with robust root and temporal direction will aid our understanding of the evolution of this fascinating nanomachine.

## Methods

### Data sources

All representative species in *Campylobacterota* with completely sequenced genomes were obtained from NCBI RefSeq database [61]. In order to cover more species diversity and also ensure genome quality, 74 complete genomes and 8 scaffold genomes were included for analyses (S2 Table). For investigation of F9 chemosensory class in the *Bacteria* domain, 2,781 representative species with complete genomes across diverse bacterial phyla were analyzed and these genome sequences was download from the NCBI genome database (https://www.ncbi.nlm.nih.gov/genome/browse#!/prokaryotes/) on Mar 9[th], 2021.

### Bioinformatics software

The phylogenetic framework for the *Campylobacterota* phylum in Figs 1A and 2A and 6A was adopted from Figs 1A and S1 in [22]. For newick tree file, see S1 Data. Other phylogenetic trees based on chemosensory or flagellar proteins were constructed by either MEGA X or Fast Tree as specified below [62, 63]. For analysis of the chemosensory and flagellar components, protein homologs were identified by blast-2.12.0 with default parameters [64]; multiple

sequence alignments were performed by MAFFT program using the e-ins-i algorithm [65]; protein domain organizations were analyzed by SMART and HHpred [66, 67]. Tree Collapse was used to produce the final tree figures with large dataset in Figs 2B and 5A and 5C [68]. Sequence logos for multiple sequence alignments in Fig 5B were generated using Weblogo (S7 Data) [69].

## Analyses of chemosensory system in *Campylobacterota*

The genes encoding components of the chemosensory system (CheA, CheW, CheV, CheB, CheR, CheC, CheD, CheX, CheY, CheZ and all chemoreceptors) in the selected genomes were identified by BLAST and the hits with e-value less than $e^{-5}$ was assigned as candidates. All the candidates were reexamined for their domain organization by searching at SMART, HHpred, and MiST 3.0 databases manually [66, 67, 70]. All chemosensory proteins except chemoreceptors are listed in S3 Table.

The genes encoding chemosensory proteins and their neighborhood genes were manually mapped on the genome and assigned with F classes according to Wuichet and Zhulin [2] _ENREF_15. Among all chemosensory classes identified in *Campylobacterota*, the F7 class has two types of gene orders, named as F7_*Arcobacter* and F7_*Sulfurospirillum* according to their gene order from the *Arcobacter* genus and *Sulfurospirillum/Sulfuricurvum/Sulfurimonas* genera.

The concatenated sequences of aligned F3 class components-CheY, CheZ and CheX were used to generate a phylogenetic tree by MEGA X using WAG model with bootstrapping 100 replications (S2 Data) [62]. The phylogenetic trees based on all CheY homologs from the *Campylobacterota* phylum (Fig 5A) was constructed by FastTree using JTT+CAT model and bootstrap analysis of 1000 replications (S5 Data) [63].

The chemoreceptors were classified into different H types based on the number of helical heptads in their signaling domains [33]. All chemoreceptors types in each genome were reexamined by MiST 3.0 database [70]. The classification of chemoreceptors was summarized in S4 Table.

## Analyses of flagellar components in *Campylobacterota*

**To identify all flagellar component in** 82 *Campylobacterota* genomes, **56 flagellar proteins** were identified by BLAST, including 9 proteins of flagellar secretion system—FlhA, FlhB, FlhX, FliH, FliI, FliO, FliP, FliQ, FliR; severn proteins of the central rod-FliF, FlgB, FlgC, FliE, FlgF, FlgG, FlgJ; seven proteins of the motor- MotA, MotB, FliG, FliL, FliM, FliN, FliY; two proteins of L- and P-rings-FlgH, FlgI; six proteins of the hook and filament-FlgD, FlgE, FlgK, FlgL, FliD, FlaG; five chaperons-FlgA, FlgN, FliJ, FliS, FliT; seven regulators-RpoN, FlgR, FlgS, FliA, FlgM, CsrA, FliW; three other regulators-FlhF, FlhG, FliK; ten additional components in *Campylobacterota*-PflA, PflB, FlgP, FlgQ, FlgV, FlgW/SwrB, FlgX, FlgY/MotE, FlgO, FlgO' (FlgO paralog). Except FlgO and FlgO' from *Arcobacter butzleri* JV22, all the other flagellar proteins can be found in *Campylobacter jejuni* 81–176, which were used as probes to perform BLAST searches. The candidates were reexamined by domain organization using SMART database and structural similarity using HHpred, and further confirmed by their gene neighborhoods. All identified protein components were summarized in S8 Table. The flagellar genes in each genome were mapped and visualize on the linearized chromosomes by RIdeogram [71].

To examine the phylogenetic relationship of flagellar components among *Arcobacter*, *Nitratiruptor* and other *Campylobacterota* species, 9 universally conserved flagellar proteins (FliE, FliF, FliG, FlgB, FlgC, FlgK, FlgL, FlhA and FlhB) were collected from 145 bacterial species

covering 15 phyla (S9 Table). Concatenated alignments of these 9 protein sequences were used to build a maximum likelihood tree by FastTree using JTT+CAT model and bootstrap analysis of 1000 replications (S3 Data) [63]. The phylogenetic trees based on FliM (Fig 5C) was constructed using the same method (S6 Data).

### Analyses of the chemosensory F9 class in *Bacteria*

2,781 representative species with complete genomes were used to assess the distribution of F9 class in the *Bacteria* domain. Two signature components of F9 class-DosM and CheW$_3$ were identified by BLAST, and their domain organization was further confirmed by SMART databases. The distribution and detailed gene order/components of F9 class were listed in S5 and S6 Tables. The concatenated sequences of aligned DosM and CheW$_3$ was used to generate a phylogenetic tree by MEGA X using WAG model with bootstrapping 100 replications (S4 Data) [62]. The tree was finally modified by iTol (http://itol.embl.de/about.cgi).

## Supporting information

**S1 Fig. The conserved gene order of F3 chemosensory class genes *cheX*, *cheY*, and *cheZ*.** (PDF)

**S2 Fig. Sequence alignment of potential CheX homologs in *Campylobacterota* species with experimentally confirmed CheX from *Thermotoga maritima* shown on the top.** (PDF)

**S3 Fig. Pairwise sequence and secondary structural comparison of CheX from *Campylobacter jejuni* and *Thermotoga maritima*.** (PDF)

**S4 Fig. Genomic distribution of flagellar genes in *Campylobacterota* species.** (PDF)

**S5 Fig. The gene order of F14 chemosensory class and flagellar locus in species of *Nitratiruptor* (*Campylobacterota*) and *Persephonella* (*Aquificota*).** (PDF)

**S6 Fig.** (A) Ternary comparison of flagellar gene distribution of six *Arcobacter* species with deep-branching species (*Lebetimonas* sp. JS138) and a close relative (*Sulfurimonas gotlandica* GD1). (B) The gene order of two big flagellar gene clusters (orange arrow) in *Arcobacter* species. (PDF)

**S7 Fig. Flagellar gene arrangement of all *Arcobacter* species.** (PDF)

**S8 Fig. The conserved gene order of *flgV*, *flgX* and *flgY* in *Campylobacterota* genomes.** (PDF)

**S9 Fig. The conserved gene order of *flgO* and its potential paralog *flgO'* in *Arcobacter* genomes.** (PDF)

**S10 Fig. Sequence alignment of FlgO and FlgO' homologs in *Arcobacter* species and *V. cholera*.** (PDF)

**S1 Table. F classes and corresponding chemosensory arrays visualized by Cyro-EM.**
(XLSX)

**S2 Table. Summary of analyzed species and their characteristics including the presence of flagellar filament, motility and the presence of full flagellar gene set.**
(XLSX)

**S3 Table. Summary of Chemosensory classes and their components in *Campylobacterota* genomes.**
(XLSX)

**S4 Table. Summary of chemoreceptor types in *Campylobacterota* genomes.**
(XLSX)

**S5 Table. Distribution of full and partial F9 class gene set in various taxon (phyla or classes).**
(XLSX)

**S6 Table. The chemosensory F9 class components in 72 species.**
(XLSX)

**S7 Table. Summary of CheY homologs in *Campylobacterota* genomes.**
(XLSX)

**S8 Table. Flagellar components in *Campylobacterota* genomes.**
(XLSX)

**S9 Table. Flagellar proteins used for phylogenetic tree construction in Fig 2B.**
(XLSX)

**S1 Data. Phylogenetic tree in Figs 1A and 2A and 6A in Newick format.**
(NWK)

**S2 Data. Phylogenetic tree in Fig 1C in Newick format.**
(NWK)

**S3 Data. Phylogenetic tree in Fig 2B in Newick format.**
(NWK)

**S4 Data. Phylogenetic tree in Fig 4B in Newick format.**
(NWK)

**S5 Data. Phylogenetic tree in Fig 5A in Newick format.**
(NWK)

**S6 Data. Phylogenetic tree in Fig 5C in Newick format.**
(NWK)

**S7 Data. Multiple sequence alignments of CheY homologs in Fig 5B in Fasta format.**
(FASTA)

## Acknowledgments

The authors would like to thank Dr. Davi Ortega for critical reading and comments of the manuscript.

## Author Contributions

**Conceptualization:** Beile Gao.

**Formal analysis:** Ran Mo, Siqi Zhu, Yuanyuan Chen, Yuqian Li.

**Funding acquisition:** Beile Gao.

**Investigation:** Ran Mo, Siqi Zhu, Yuanyuan Chen, Yuqian Li, Yugeng Liu.

**Supervision:** Beile Gao.

**Validation:** Beile Gao.

**Writing – original draft:** Ran Mo, Siqi Zhu.

**Writing – review & editing:** Beile Gao.

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
