## [Decision Letter · Decision Letter 0]

9 May 2022

Dear Dr Gao,

Thank you very much for submitting your Research Article entitled 'The Evolutionary Path of Chemosensory and Flagellar Macromolecular Machines in Campylobacterota' to PLOS Genetics.

The manuscript was fully evaluated at the editorial level and by three independent peer reviewers. The reviewers appreciated the attention to an important problem, but raised some substantial concerns about the current manuscript. Based on the reviews, we will not be able to accept this version of the manuscript, but we would be willing to review a much-revised version. We cannot, of course, promise publication at that time.

If you decide to revise the manuscript for further consideration at PLOS Genetics, please aim to resubmit within the next 60 days, unless it will take extra time to address the concerns of the reviewers, in which case we would appreciate an expected resubmission date by email to plosgenetics@plos.org.

[LINK]

We are sorry that we cannot be more positive about your manuscript at this stage. Please do not hesitate to contact us if you have any concerns or questions.

Yours sincerely,

Xavier Didelot

Associate Editor

PLOS Genetics

Josep Casadesús

Section Editor: Prokaryotic Genetics

PLOS Genetics

Reviewer's Responses to Questions

**Comments to the Authors:**

Reviewer #1: The manuscript by Mo et al performs a phylogenetic analysis on the Campylobacterota clade of bacteria and subsequently focus on the flagellar and chemotaxis homologs encoded in the genomes. I think they show that the most ancestral members of the clade have a flagellum whose genes are dispersed throughout the genome and a particular chemotaxis system type “F3”. As one moves deeper in the tree, there are relatives in which all of the flagellar and chemotaxis genes are lost and later relatives reacquire a contiguous set of flagellar genes by horizontal transfer and the chemotaxis system that accompanies the change also is of a different type. There is an over-abundance of details presented, most of which meant essentially nothing to me, but the above is my take-home message (which may or may not be correct). The support for the flagellar substitution made sense to me but I really didn’t understand the relevance of the different chemotaxis system types. Perhaps it would help to describe what goes into the definition of the different chemotaxis classes and why/how those classes are different or would matter (something about F9 being soluble but I didn’t get much out of that). Maybe start with the idea that the flagellum changes and then describe the chemotaxis changes? As it stands now, the chemotaxis differences are presented first and later we learn the flagellum changes so the former makes sense because the chemotaxis systems are subordinate. Whatever the case, PLoS Genetics is a generalist journal and some greater effort should go into making the story generalist accessible, this is particularly important because I don’t totally understand what is going on here and I’m a specialist in the field.

Line 54. “It” I assume the “it” is the flagellum but please clarify. Moreover the bacterial flagellum is rather narrowly distributed in the bacterial lineage and type IV pilus mediated twitching motility and divergent mechanisms of gliding are almost certainly more widespread than flagellar mediated movement. Finally, the archaellum of the archaea is not related to the bacterial flagellum. Reword.

Line 59. “chemosensory system is evolutionarily dynamic in contrast to other highly conserved macromolecular machines” I don’t think this is true. I think the chemotaxis proteins are more highly conserved than the machines they control. For example, chemotaxis protein homologs control both the flagellum and the pilus of Pseudomonas aeruginosa and are more similar to each other than the flagella are to pili. Lines 77-79 support my concern.

Para starting 128 seems like introductory material. Or rather, seems to contain no new information.

Line 140 mentions CheA with an “additional receiver domain” but it is never mentioned how many receiver domains most CheA proteins have.

Section starting 139. I don’t understand what this paragraph is trying to tell me. I think it is justification for calling the Campylobacter chemotaxis system an F3 system (I don’t know what this means either) but all of the information seems inferential and again previously reported. Perhaps better for introduction. Clarify the point of this paragraph. In general, I often found myself uncertain as to what information was new and what was previously published.

Line 158. Check syntax. The gene organization is described as unusual as they genes do not seem related but then the second clause says that it isn’t unusual to have genes organized like this? Clarify.

Line 190. What is the H nomenclature? E.g. 36H, 44H etc?

Line 471. The statement suggests that Arcobacter specifically lost structural components but from my reading the manuscript, I thought the story was that all of the flagellar genes were lost and then Arcobacter reacquires a different complete set of flagellar genes by horizontal gene transfer. Thus, it didn’t actually lose certain (high torque) components in particular, it just reacquired a (low torque) flagellar system that doesn't require, and never had them.

Reviewer #2: This manuscript by Mo, et al reported the evolution of bacterial flagella and its chemosensory systems in the Campylobacterota phylum based on the sequence analysis. Although the manuscript presents a large amount of data, it is difficult to extract informative conclusions from the current dataset. As I described the below, the extent of this issue indicates that a simple revision may not be sufficient to raise the paper to the level required for publication.

A major concern is an apparent discrepancy between the results of this study and previously published work. The authors insist on analyzing the signal transduction system previously categorized as F3 class. The gene set of F3 class does not have the gene of cheX at the original definition in ref 2. However, this gene is including for their analyses in Fig 1 and elsewhere. I am concerned whether these analyses were performed correctly.

The Introduction part described a general discussion of other bacterial species, and there is little description of the bacteria belong to Campylobacteriota phylum. It is difficult to follow the line or arguments what is a fundamental breakthrough in the research area in this phylum.

The first paragraph of Results and Discussion is a summery of ref 24, which is an authors’ preprint in bioRxiv. It is unclear why the authors include this section as results.

Results and Discussions part is not always convincing and are often overinterpreted. Authors analyzed the flagellar genes in Nitratifractor and Arcobacterin Fig 2-5, but the motility and chemotaxis of these bacteria have not been well understood whether these gene are functional. The authors’ conclusions are not well-founded.

The data presented here are merely correlative and do not show causal relationship for the flagellar structure and its torque generation. The authors’ data are just sequence analyses. The precise measurements of the torque of flagellar motors are only restricted in few species, and the similar discussion about the motor structure have been already reported in ref 52. In addition, the images in Fig 6B are not original data of the author. This is a diversion of another authors’ work from Movie S2 in ref 53, and Fig 2 in ref 26.

Reviewer #3: This paper describes a thorough and technically sound analysis of chemosensory and flagellar systems in Campylobacterota, one of the many bacterial phyla.

Concerns:

1. English must be improved.

2. Multiple sequence alignments (e.g. in Fasta format) and phylogenetic trees (e.g. in Newick format) must be provided as supplementary files. At least the key ones.

3. Line 494: How the representative set of 2,781 species was selected? I cannot find information about this set anywhere in the supplement.

**Have all data underlying the figures and results presented in the manuscript been provided?**

Reviewer #1: Yes

Reviewer #2: Yes

Reviewer #3: **No: **Multiple sequence alignments (e.g. in Fasta format) and phylogenetic trees (e.g. in Newick format) must be provided as supplementary files. At least the key ones.

PLOS authors have the option to publish the peer review history of their article (what does this mean?). If published, this will include your full peer review and any attached files.

Reviewer #1: **Yes: **Daniel Kearns

Reviewer #2: No

Reviewer #3: No

---

## [Decision Letter · Decision Letter 1]

27 Jun 2022

Dear Dr Gao,

We are pleased to inform you that your manuscript entitled "The Evolutionary Path of Chemosensory and Flagellar Macromolecular Machines in Campylobacterota" has been editorially accepted for publication in PLOS Genetics. Congratulations!

Yours sincerely,

Xavier Didelot

Associate Editor

PLOS Genetics

Josep Casadesús

Section Editor: Prokaryotic Genetics

PLOS Genetics

Comments from the reviewers (if applicable):

Reviewer's Responses to Questions

**Comments to the Authors:**

Reviewer #1: The authors have addressed my concerns.

**Have all data underlying the figures and results presented in the manuscript been provided?**

Reviewer #1: Yes

PLOS authors have the option to publish the peer review history of their article (what does this mean?). If published, this will include your full peer review and any attached files.

Reviewer #1: **Yes: **Daniel Kearns

**Data Deposition**

http://datadryad.org/submit?journalID=pgenetics&manu=PGENETICS-D-22-00425R1

**Press Queries**

---

## [Editor Report · Acceptance letter]

8 Jul 2022

PGENETICS-D-22-00425R1 

The Evolutionary Path of Chemosensory and Flagellar Macromolecular Machines in Campylobacterota 

Dear Dr Gao, 

We are pleased to inform you that your manuscript entitled "The Evolutionary Path of Chemosensory and Flagellar Macromolecular Machines in Campylobacterota" has been formally accepted for publication in PLOS Genetics! Your manuscript is now with our production department and you will be notified of the publication date in due course.

With kind regards,

Marianna Bach

PLOS Genetics

On behalf of:
